# Sub-1 Volt and high-bandwidth visible to near-infrared electro-optic modulators

**Dylan Renaud** [1,5] ✉, **Daniel Rimoli Assumpcao**[1,5], **Graham Joe**[1], **Amirhassan Shams-Ansari** [1], **Di Zhu** [1,2], **Yaowen Hu**[1,3], **Neil Sinclair**[1,4] & **Marko Loncar** [1] ✉

Integrated electro-optic (EO) modulators are fundamental photonics components with utility in domains ranging from digital communications to quantum information processing. At telecommunication wavelengths, thin-film lithium niobate modulators exhibit state-of-the-art performance in voltage-length product ($V_\pi L$), optical loss, and EO bandwidth. However, applications in optical imaging, optogenetics, and quantum science generally require devices operating in the visible-to-near-infrared (VNIR) wavelength range. Here, we realize VNIR amplitude and phase modulators featuring $V_\pi L$'s of sub-1 V · cm, low optical loss, and high bandwidth EO response. Our Mach-Zehnder modulators exhibit a $V_\pi L$ as low as 0.55 V · cm at 738 nm, on-chip optical loss of ~0.7 dB/cm, and EO bandwidths in excess of 35 GHz. Furthermore, we highlight the opportunities these high-performance modulators offer by demonstrating integrated EO frequency combs operating at VNIR wavelengths, with over 50 lines and tunable spacing, and frequency shifting of pulsed light beyond its intrinsic bandwidth (up to 7x Fourier limit) by an EO shearing method.

Integrated photonics at visible and near-infrared (VNIR) wavelengths is important for applications ranging from sensing[1–3] and spectroscopy[4] to communications[5] and quantum information processing[6,7]. For example, visible integrated photonic platforms can be combined with any of the large variety of atomic or atomic-like systems with transitions in the VNIR such as alkali and alkaline-earth metal atoms[8–10], rare-earth ions[11], diamond color centers[12,13] and quantum dots[14–16]. Concerning quantum applications, VNIR photonics enables photon routing[17,18], spectral shifting for interfacing disparate quantum emitters[11,19], or realizing higher-dimensional encoded quantum states[20,21], all in a scalable and compact approach.

A variety of visible integrated photonic platforms have been demonstrated, including silicon nitride[2,22–25], aluminum ntiride[26,27], diamond[28,29], and lithium niobate (LN)[30–32]. LN is particularly compelling due to its large electro-optic (EO) coefficient, low optical loss, and wide transparency window, making it the workhorse material for the modern day telecommunications industry. Recent work has shown the promise of thin-film lithium niobate (TFLN) at telecommunications wavelengths[33]. Beyond the inherent miniaturization and integratability achievable with TFLN, the strong optical confinement and increased tailorability have enabled performance not achievable with bulk LN, including CMOS compatible drive voltages and high bandwidth operation[34–36]. As a result of LN's large transparency window, EO TFLN devices in the visible regime have been demonstrated[30–32]. However, half-wave voltages ($V_\pi$) and large bandwidths beyond that realized in visible bulk devices has yet to be demonstrated in VNIR TFLN. In particular, the combination of high-bandwidth and low drive-voltage optical modulation would enable on-chip routing and spectral control: a critical requirement for quantum applications.

In this work, we realize VNIR TFLN amplitude and phase modulators (Fig. 1a) operating with $V_\pi L$ of sub-1 V · cm (Fig. 1b), extinction ratios beyond 20 dB, and 3 dB EO bandwidths in excess of 35 GHz.

[1]John A. Paulson School of Engineering and Applied Sciences, Harvard University, Cambridge 02139 MA, USA. [2]Institute of Materials Research and Engineering, Agency for Science, Technology and Research (A*STAR), Singapore 138634, Singapore. [3]Department of Physics, Harvard University, Cambridge 02139 MA, USA. [4]Division of Physics, Mathematics and Astronomy, and Alliance for Quantum Technologies (AQT), California Institute of Technology, Pasadena 91125 MA, USA. [5]These authors contributed equally: Dylan Renaud, Daniel Rimoli Assumpcao. ✉e-mail: renaud@g.harvard.edu; loncar@g.harvard.edu

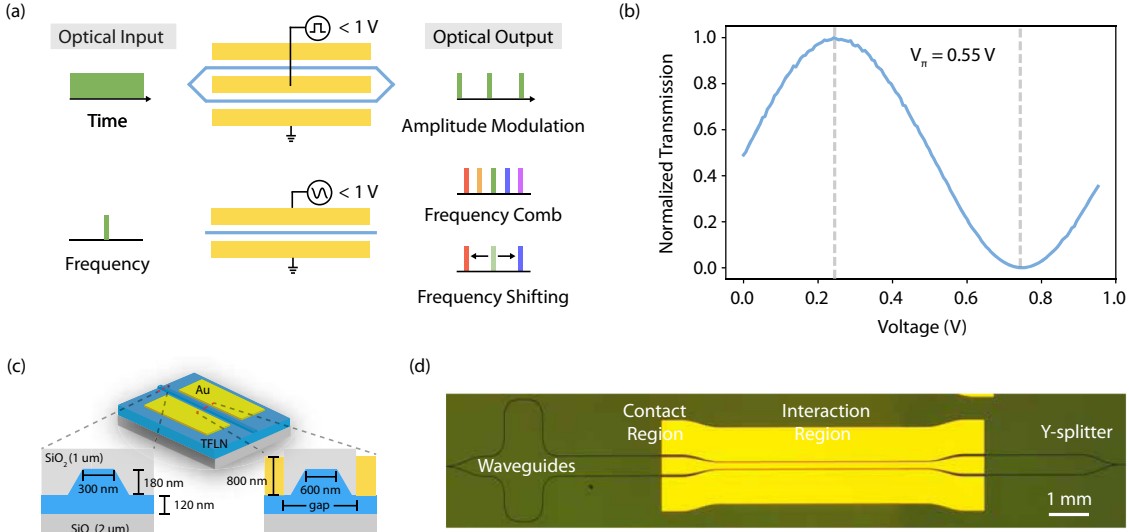

**Fig. 1 | Ultra-low $V_\pi$ modulators operating at visible-to-near-infrared wavelengths. a** In the time domain, VNIR amplitude modulators with ultra-low drive voltages (< 1 V) can modulate continuous-wave optical inputs at CMOS voltages. Similarly, sub-volt phase modulators enable VNIR frequency comb generation and frequency shifting over multiple pulse bandwidths. **b** Normalized optical transmission of a 10 mm long amplitude (Mach-Zehnder) modulator as a function of the applied voltage. At $\lambda = 738$ nm, the $V_\pi$ is 0.55 V at 1 MHz. **c** Cross section illustrations of modulator waveguide and electrode regions. **d** Optical micrograph of a 5 mm long VNIR TFLN amplitude modulator. The micrograph additionally shows the unbalanced amplitude modulator waveguides, along with the Y-splitters, probe contact region of the electrodes, and the electrode gap taper along the probe contact region to the interaction region.

We perform two demonstrations to highlight applications of these devices. We demonstrate an integrated and tunable EO frequency comb source in the VNIR, showing over 50 lines in a single comb at 638, 738, and 838 nm, and displaying flat-top spectra with less than 10 dB power variation. Furthermore, we use our devices to demonstrate spectral shearing of optical pulses over 7 times their intrinsic spectral bandwidth. Together these demonstrations highlight the widespread utility of TFLN modulators operating in the VNIR spectrum.

## Results
### Design and fabrication
Figure 1c illustrates the design of our TFLN VNIR modulators. We fabricate devices on 300 nm thick X-cut TFLN on 2 μm of thermally grown silicon dioxide on Si (NanoLN). For complete device fabrication details, see methods. Outside of the electrode region, the waveguides are designed to be single mode (support transverse-electric, $TE_{00}$, and transverse-magnetic, $TM_{00}$) at 740 nm. We choose this constraint to minimize excitation of higher-order modes, which can lead to a reduction in EO performance in the electrode region. Using finite-difference eigenmode simulations (Lumerical), the required waveguide top width for single mode operation is determined to be approximately 300 nm. A disadvantage of this width is that it reduces mode confinement. This leads to higher optical loss due to mode overlap with sidewalls and cladding, and absorption loss from electrodes. For this reason, we adiabatically increase the waveguide width to 600 nm in the electrode region. Finally, our amplitude modulators feature Y-splitters with excess losses of approximately 0.2 dB/splitter[30]. An optical micrograph of a fabricated 5 mm long amplitude modulator is provided in Fig. 1d.

For the electrodes, we employ a push-pull configuration with coplanar waveguide (CPW) travelling-wave electrodes. Finite element method (COMSOL) simulations are used to design electrodes with impedance close to 50 Ω and a simulated microwave phase index of $n_{RF} = 2.22$ at 50 GHz. Due to the relatively large optical group index at VNIR wavelengths compared to telecommunication wavelengths ($n_{vis} \approx 2.38$, $n_{tel} \approx 2.25$), perfect velocity matching requires a reduction in bottom oxide (BOX) thickness and/or gold thickness, which comes at the expense of increased optical and RF loss. To avoid this, our devices have an index mismatch between the microwave phase and optical group index of the $TE_{00}$ mode of $\Delta n$-0.17. For an impedance matched, 1 cm long lossless modulator, this index mismatch corresponds to a theoretical bandwidth of ~80 GHz.

### Visible-to-near-infrared Mach Zehnder modulators
We fabricate 1 cm long Mach-Zehnder modulators (MZMs) with varying gap sizes and experimentally evaluate their performance across both wavelength and electrode gap parameter spaces. The experimental setup is shown in the inset of Fig. 2a (see methods for measurement details).

As shown in Fig. 2a, the $V_\pi$ of our 1 cm long, 3 μm gap devices is as low as 0.42 V at 532 nm, and increases only slightly to 0.45, 0.55, and 0.85 V at 638, 738, and 838 nm, respectively. The increase in $V_\pi L$ for longer wavelengths follows from the smaller phase accumulation for the same modulator length. Our $V_\pi L$ is a factor of 2-3 smaller, depending on wavelength considered, than the best previously reported values for VNIR TFLN modulators, without compromising bandwidth or device insertion loss[30−32]. We note that our improvement stems predominantly from the reduction in electrode gap, i.e., enhancement in optical-microwave field overlap.

We perform the same measurements at 738 nm, but with MZMs of varied gap sizes, seeing an increasing $V_\pi L$ for larger gap sizes (Fig. 2b). For comparison, we also theoretically calculate $V_\pi L$ as a function of gap. The simulated response shows excellent agreement with our measured results. Notably, we measure $V_\pi L < 1$ V · cm for gap sizes as large as 5 μm. For comparison, recent work on VNIR devices with smaller gaps (2 μm) have reported larger low frequency $V_\pi L$[31].

We extract the on-chip modulator loss by fabricating and measuring 3 μm gap modulators of varying electrode length. From this we obtain an on-chip loss of ~ 0.7 ± 0.2 dB/cm (see supplementary fig. 1 for details). This value is over an order of magnitude smaller than that reported in other recent demonstrations for VNIR LN modulators[32,37]. Including the lensed fiber-to-chip coupling loss (~ 7 dB/facet), the total device insertion loss comes to ~ 15 dB. We note that because the total device insertion loss is dominated by coupling loss (mismatch between the lensed fiber and rib waveguide mode), it can be reduced by nearly an order of magnitude using techniques such as tapered fiber

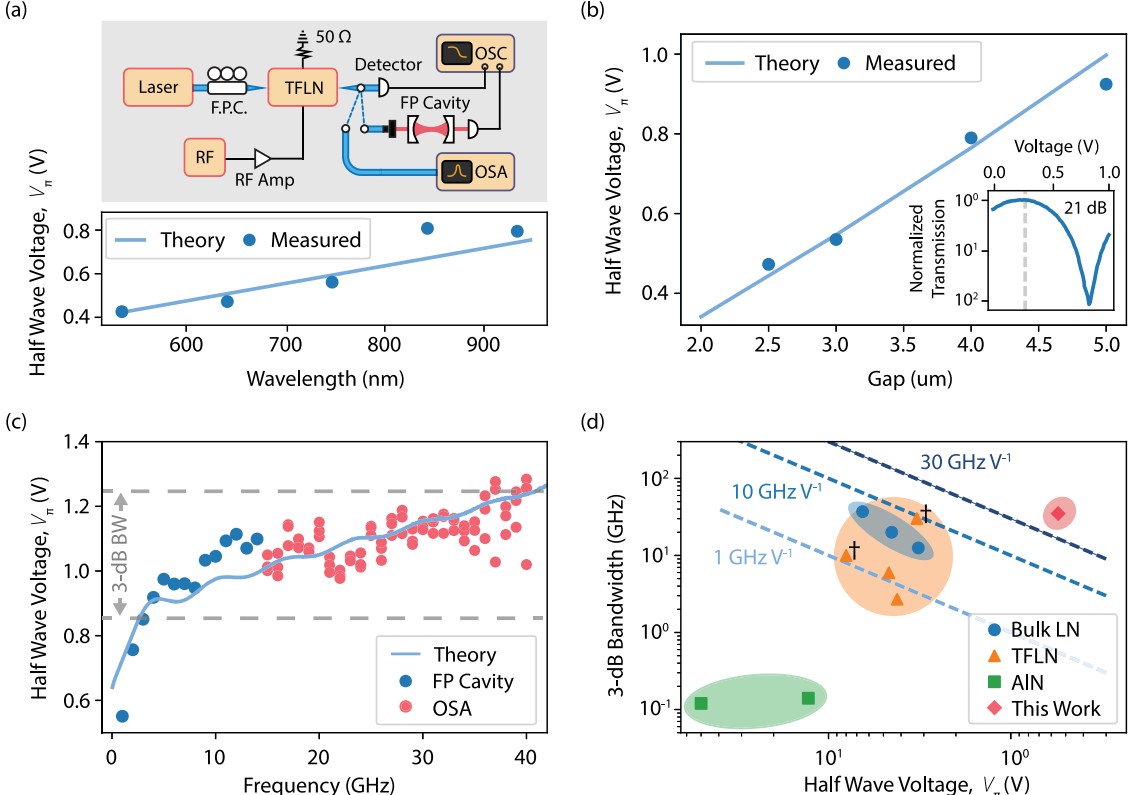

**Fig. 2 | Ultra-low $V_\pi$ visible-to-near-infrared wavelength Mach-Zehnder modulators with greater than 35 GHz bandwidth. a** Experimental setup illustration and measured low-frequency (1 MHz) $V_\pi$ for a 1 cm MZM with an electrode gap of 3 μm. Data shown corresponds to $V_\pi$ at $\lambda$ = 532, 638, 738, 838, and 938 nm. Simulated $V_\pi$ is shown by the solid line. **b** Simulated and measured $V_\pi$ (1 MHz) at $\lambda$ = 738 nm for varied electrode gap. Inset shows a measured extinction ratio of ~21 dB for a 1 cm long, 3 μm gap modulator. **c** Frequency dependence of $V_\pi$ for a 1 cm long MZM. A 3 dB EO bandwidth of ~35 GHz is extracted from the response. The bandwidth is measured with respect to a low frequency reference, here taken to be 3 GHz. The grey dashed-lines denote the 3 dB bandwidth w.r.t 3 GHz. **d** Comparison of modulator figure of merit BW/$V_\pi$ between this work, state-of-the-

art commercial LN modulators, previous VNIR thin-film LN modulators, and other VNIR modulator platforms. This work exhibits significantly higher BW/$V_\pi$ values than all previously reported works, including previously demonstrated TFLN VNIR modulators[30–32,37]. The dashed lines correspond to constant values of BW/$V_\pi$. Note that for fair comparison, we compare the reported $V_\pi$ at <1 GHz for all devices. The TFLN data points with cross annotations denote devices for which the reported BW was limited by the equipment used. Detailed comparison of referenced works can be found in supplementary tables 1 and 2. RF Radiofrequency, F.P.C. Fiber polarization controller, TFLN Thin-film lithium niobate, OSC Oscilloscope, OSA Optical Spectrum Analyzer, FP Fabry-Perot.

coupling[38]. In addition, we observe an extinction ratio of over 25 dB for 5 μm gap devices, and ~21 dB for 3 μm gap devices (Fig. 2b, inset), comparable with state-of-the-art.

To evaluate the high frequency response of the devices, the 3 dB EO bandwidth of our MZM is extracted via first measuring the optical frequency spectrum of the transmitted light on a Fabry-Perot (FP) cavity or optical spectrum analyzer (OSA) and fitting the sideband powers to a Bessel function for varied frequency applied sinusoidal voltage. Figure 2c shows the high frequency response for a 3 μm gap, 1 cm long MZM operating at 738 nm. The extracted 3 dB bandwidth is approximately 35 GHz (w.r.t. to 3 GHz), and is limited by RF loss of the CPW (1.35 dB cm$^{-1}$ GHz$^{-1/2}$). A non-DC reference is chosen due to both the rapid roll-off originating from the CPW impedence mismatch, and the commonly observed instability in LN modulators at low frequencies due to photorefractive effects. We use the measured electrical transmission coefficients of the CPW to also theoretically predict the modulator 3 dB bandwidth, which we find to be ~36 GHz, in excellent agreement with our measured result. We emphasize that since the device bandwidth is limited by CPW RF loss, it can be improved upon by implementing capacitively loaded traveling wave electrodes to reduce current crowding and its associated increase of RF loss[39].

To fully contextualize the performance of our device, we compare our results with other previously demonstrated VNIR modulators in

terms of bandwidth and low-frequency $V_\pi$ (BW/$V_\pi$ ratio), including both integrated and non-integrated VNIR modulators (Fig. 2d). We emphasize the improvement in performance between this work and state-of-the-art commercial bulk VNIR LN modulators. While prior works have demonstrated the superior performance of TFLN over bulk LN in the telecommunication band[40], to date, the same has not been shown in TFLN VNIR modulators. Here, we show that VNIR TFLN modulators can exhibit voltage-bandwidth performance exceeding 30 GHz V$^{-1}$, a performance metric not achievable in bulk devices, or other current integrated VNIR photonic platforms. Finally, we note that while other visible modulator platforms have recently been demonstrated[41,42], they are presently limited to operating frequencies in the 1–100 kHz range.

### Visible-to-near-infrared electro-optic frequency comb
To demonstrate the utility of these EO devices for sensing applications, we fabricate TFLN phase modulators (PM) operating at VNIR wavelengths. Because of the low required driving voltages and broadband optical operation, we use these devices to generate EO frequency combs operating at high-frequency and with variable comb spacing. Our combs feature over 50 sidebands when driven at 30 GHz by a ~3 W microwave source (~8$V_\pi$), and they operate over a broad wavelength range. Results for a 3 μm device operating at 638, 738, and 838 nm are shown in Fig. 3a–c. The insets depicts the comb spacing between two

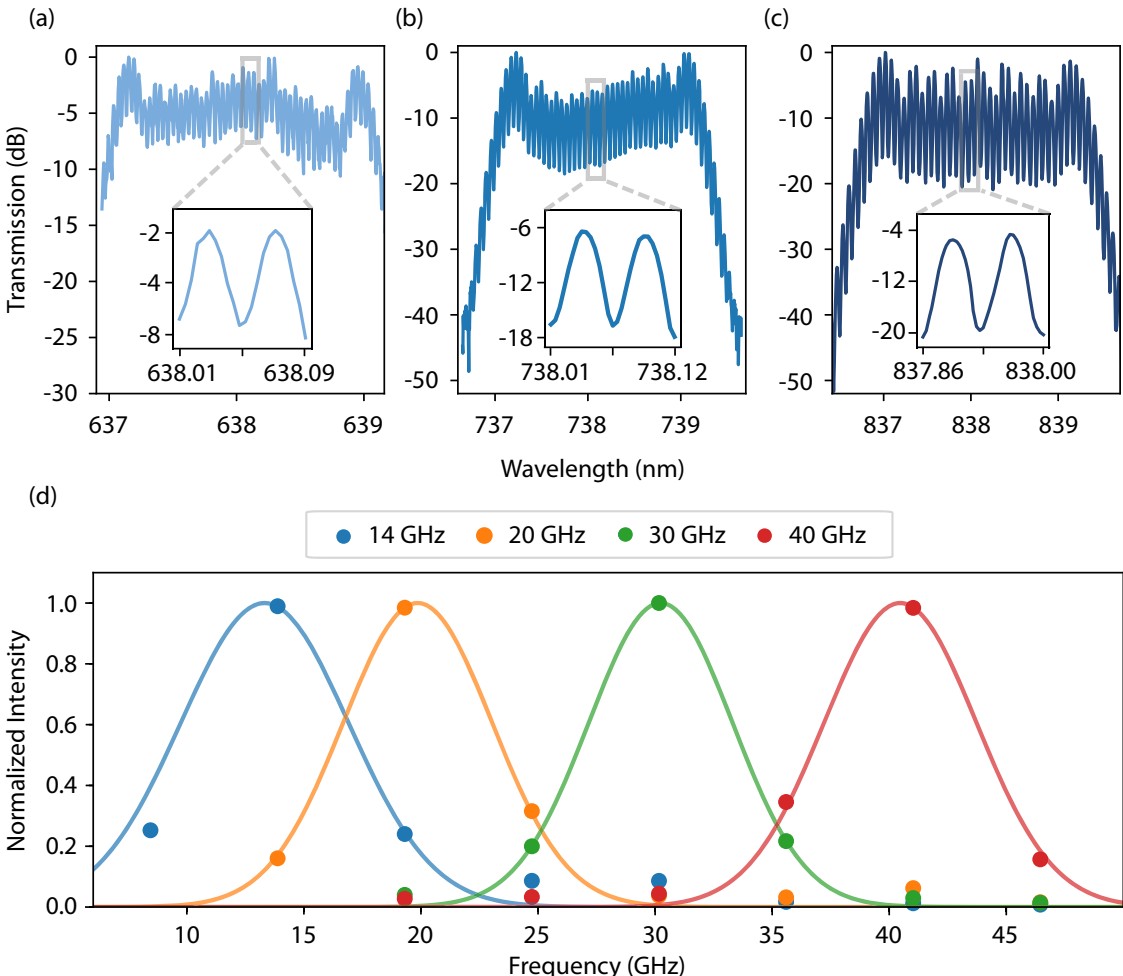

**Fig. 3 | Integrated visible-to-near-infrared electro-optic frequency combs.** Tunable frequency combs operating at **a**, $\lambda$ = 638, **b** 738 nm, and **c** 838 nm with 30 GHz spacing and more than 50 lines. Insets show magnified view of the comb lines. The asymmetry shown in the 638 nm and 738 nm combs is attributable to the waveguide supporting higher order modes at shorter wavelengths (see supplementary fig. 2). **d** Normalized 1st modulated sideband as a function of applied RF frequency showing comb tunability up to 40 GHz. The pump wavelength is 738 nm. Spectra are under-sampled due to the limited resolution of the spectrometer. Gaussian fits are provided to guide the eye.

lines at higher magnification. Due to the limited resolution of the OSA at these wavelengths, the comb visibility is not fully resolved. We further note that for shorter wavelengths, the comb envelope displays greater asymmetry. This phenomenon originates from the fact that the waveguides begin to support higher order modes at these wavelengths, each of which propagate at different group velocities and possess varying velocity mismatch with respect to the RF field. We support this assessment by calculating the theoretical comb spectrum after including additional modes which shows good qualitative agreement with our measured results (see supplementary figure 2). Finally, in Fig. 3d, the 1st sideband is shown as a function of applied RF drive frequency, thereby illustrating the tunable nature of the VNIR EO combs.

This is an important component that can be utilized for a variety of applications including sensing[43], astrophysical spectroscopy[4], and frequency-bin encoding of quantum information[20].

**Visible spectral shearing**

Our low $V_\pi$ modulators enable high-bandwidth frequency control of input light. Namely, by applying a quasi-linear phase ramp $\phi(t) = -Kt$ to the modulator, input light can be shifted in frequency by $K$. This frequency shift is referred to as spectral shearing[44,45]. This is useful for quantum applications where frequency shifting can be used to bridge

the inhomogenous distribution of quantum emitters or for performing frequency bin operations on non-classical states[46]. The former is particularly useful at visible wavelengths where a variety of quantum emitters have their optical transitions.

The shift achievable via frequency shearing is dependent on the total phase applicable to the device. This is proportional to $\frac{V}{V_\pi}f_{\mathrm{RF}}$, where $V$ is the applied voltage, and $f_{\mathrm{RF}}$ is the frequency of the applied RF tone. Thus a low $V_\pi$ phase modulator is required to achieve a large total shift. Although previous demonstrations of shearing have demonstrated large frequency shifts up to 640 GHz[47], these demonstrations relied on ultra-short pulses to utilize a high frequency RF drive[45,47,48], and thus shifting beyond the bandwidth of the pulse has not been demonstrated.

We use a 100 MHz RF tone with an amplitude of ~ $20V_\pi$ applied to our device and lock 1 ns duration square-shaped optical pulses ($\lambda$ = 737 nm) to the rising or falling linear regime of the RF tone to apply a quasi-linear phase profile to the pulse (Fig. 4a, b). Our TFLN device is a 1 cm long, 3 μm gap phase modulator with a $V_\pi$ ~ 1 V at 100 MHz. We observe a spectral shift of ± 6.6 GHz when locking the pulse to the rising or falling edge, respectively (Fig. 4c). Approximately 85% of input power is shifted into the desired lobe, with our estimate limited by the finite extinction of the input optical pulses (~ 20 dB), which can be improved via gating the detected signal.

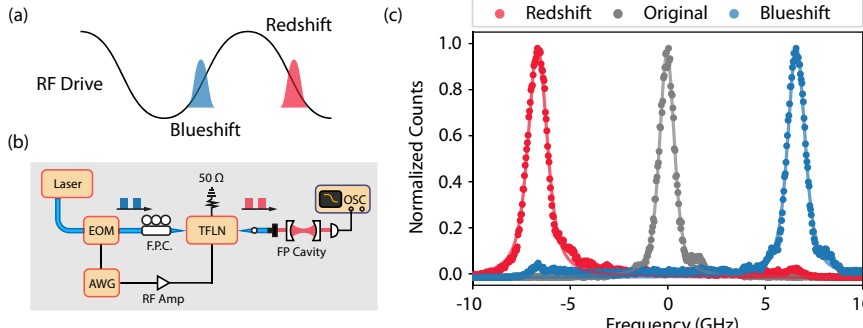

**Fig. 4 | Visible spectral shearing. a** Principle of spectral shearing, where the rising (falling) edge of a sinusoidal tone is used to provide a linear phase across an optical pulse in order to blue (red) shift its frequency. **b** Diagram of the experimental setup. An arbitrary waveform generator (AWG) is used to generate both 1 ns square electrical pulses and the 100 MHz sinusoidal drive for the phase modulator. The square electrical pulses generate square optical pulses via a commercial amplitude EO modulator, which are then routed to the low $V_\pi$ TFLN phase modulator. The RF drive from the AWG is amplified before reaching the device. **c** Frequency spectrum of the pulses after applying a sinusoidal tone and synchronizing the pulse with the falling (redshift) or rising (blueshift) edge of the RF tone, showing a shift of ± 6.6 GHz (over 7 times the pulse bandwidth) relative to the original spectrum with no RF tone applied. EOM Electro-optic modulator, AWG Analog waveform generator, F.P.C. Fiber polarization controller, TFLN Thin-film lithium niobate, FP Fabry-Perot, OSC Oscilloscope.

Given the pulse's bandwidth of ~ 0.9 GHz, the observed shift is over 7 times larger than the pulse bandwidth, almost an order of magnitude larger than previous studies have achieved[47]. We emphasize that the wavelength and pulse duration used in this work is comparable to the wavelength and lifetime of photons emitted from a variety of different visible solid-state quantum emitters and the achieved shift similar to their corresponding inhomogenous distribution, thus providing a route for deterministically bridging this frequency gap[49].

## Discussion

We have demonstrated VNIR TFLN phase and amplitude modulators featuring sub-1 V half-wave voltage, extinction ratios above 20 dB, on-chip insertion loss as low as 0.7 dB, and electro-optic bandwidths exceeding 35 GHz. With this performance we have demonstrated an integrated VNIR EO frequency comb with over 50 lines, and measured spectral shifting with shifts beyond the intrinsic bandwidth of the pulse. Together, these results show the suitability of these devices for both spatial and spectral control of input light. The performance and scalability of our integrated platform ensures its suitability for a wide variety of applications. Through combining this visible platform with other demonstrated TFLN technologies such as periodically poled lithium niobate (PPLN)[50] or laser integration[51,52], applications such as visible on-chip spectroscopy, photon pair generation and manipulation, and efficient visible light communication can be realized.

## Methods
### Device fabrication
The optical waveguide layer is realized using $Ar^+$-based reactive-ion etching (RIE) with a lithographically defined (Elionix ELS-F125) hydrogen silsesquioxane hard-mask[30]. After etching (180 nm) and cleaning, the device is cladded with silicon dioxide (~1 μm) via plasma-enhanced chemical vapor deposition. To reduce optical loss and mitigate photorefractive effects, devices are subsequently annealed[36,53]. Next, the electrode layer is defined with an electron beam lithography step, RIE ($C_3F_8$, $Ar^+$), electron beam evaporation (~10/800 nm, Ti/Au), and lift-off.

### Measurement details
#### Measurement of half-wave voltage.
Devices are characterized in the 634–638 nm and 720–940 nm ranges using New Focus Velocity and M2 SolsTis tunable lasers. The laser source polarization is set using a fiber polarization controller (Thorlabs, FPC560), and the output is launched

into the device coupling waveguide using a single-mode lensed fiber (OZ Optics TSMJ-3A-650-4/125-0.25-20-2-10-1). The transmitted light is then collected using a second lensed fiber at the output waveguide and sent to a high sensitivity avalanche photodetector (APD410A), home-built fabry-perot cavity (FP), or optical spectrum analyzer (OSA, AQ6730) depending on the measurement being performed. Coplanar ground-signal-ground (GSG) electrodes are contacted using 50 Ω GSG probes (GGB Industries, 40A-GSG-100-F). The DC performance is evaluated using a 1 MHz triangle waveform, and the normalized transmission is recorded as a function of applied voltage.

#### Measurement of high frequency half wave voltage.
In order to measure the half-wave voltage ($V_\pi$) of the MZMs at high frequencies, a sinusoidal RF tone is applied at varying frequencies and the resulting optical spectrum is measured. The optical frequency spectrum of an MZM given an input CW optical carrier frequency of $\omega_0$, an applied RF tone at frequency $\omega_m$ and amplitude $V_0$, and internal phase between the arms of $\phi$ is given by:

$$I(\omega_0 + k\omega_m) \propto \frac{1}{2} J_k^2(\pi V_0 / V_\pi)[1 + (-1)^k \cos(\phi)] \qquad (1)$$

where $k$ is an integer of the harmonic of the drive frequency[54].

The frequency spectrum is measured using a home-built Fabry-Perot cavity (linewidth = 200 MHz) for lower frequencies (< 15 GHz) and an optical spectrum analyzer (OSA, AQ6730) or Czerny-Turner spectrometer (Princeton Instruments, SpectraPro HRS) for higher frequencies. For the FP cavity, the intensity of the carrier is measured as a function of applied voltage and fit to (1). For measurements with the OSA, a wider spectrum featuring multiple sidebands is measured at various powers and the relative intensities of the even sidebands of each spectrum is fit to equation (1).

#### Frequency shearing measurement.
To implement spectral shearing, the RF tone applied to the device and optical pulse are generated using the same arbitrary waveform generator (AWG, Tektronix 700001b), while also ensuring minimal jitter between the two. Optical pulses are defined using a commercial amplitude electro-optic modulator (EOSpace, AZ-AV5-40-PFA-PFA-737) with a continuous-wave laser. The spectrum of the pulse is measured using a FP cavity. We numerically find the non-linearity of the sine tone in this quasi-linear region to have a negligible effect on the resulting spectrum for the given drive frequency and pulse duration.

## Data availability

All data that supports the conclusions of this study are included in the article and the Supplementary Information file. The data presented in this study is available from the corresponding authors upon request.

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

## Acknowledgements

This work was supported in part by AFOSR FA9550-20-1-0105 (M.L.), FA9550-19-1-0376 (M.L.), ARO MURI W911NF1810432 (M.L.), NSF EEC-1941583 (M.L.), OMA-2137723 (M.L.), and OMA-2138068 (M.L.), DOE DE-SC0020376 (M.L.), MIT Lincoln Lab 7000514813 (M.L.), AWS Center for Quantum Networking's research alliance with the Harvard Quantum Initiative (M.L.), Ford Foundation Fellowship, (D.R.), NSF GRFP No. DGE1745303 (D.R., D.A.), NSERC PGSD scholarship (G.J.), Harvard Quantum Initiative (HQI) postdoctoral fellowship (D.Z.), A*STAR Central Research Fund (D.Z.), AQT Intelligent Quantum Networks and Technologies (N.S.), and NSF Center for Integrated Quantum Materials No. DMR-1231319 (D.R., N.S.). We acknowledge fruitful discussions with Lingyan He, Prashanta Kharel, Ben Dixon, and Alex Zhang. Device fabrication was performed at the Center for Nanoscale Systems (CNS), a member of the National Nanotechnology Coordinated Infrastructure Network (NNCI), which is supported by the National Science Foundation under NSF Grant No. 1541959.

## Author contributions

G.J. and D.R. designed devices. D.R. fabricated devices. D.A., D.R., and G.J. designed and performed the measurements. A.S. assisted with electronics measurements. D.A., D.R., and D.Z. analyzed the data. Y.H. provided a theory on frequency combs. D.R., D.A., and A.S. wrote the manuscript with extensive input from the other authors. M.L. and N.S. supervised the project. These authors contributed equally: D.R. and D.A.

## Competing interests

M.L. is involved in developing lithium niobate technologies at Hyper-Light Corporation. The remaining authors declare no competing interests.
