## [Peer Review File · Nature Communications]

Sub-1 Volt and High-Bandwidth Visible to Near-Infrared
Electro-Optic ModulatorsREVIEWER COMMENTS

Reviewer #1 (Remarks to the Author):

Dear authors and editor,

I have carefully read through the manuscript entitled, “Sub-1 Volt and High-Bandwidth Visible to Near-Infrared Electro-Optic Modulators,” by Dylan Renaud et al. The authors present experimental results that demonstrate high-bandwidth electro-optic modulators (EOMs) for visible and near-infrared light with a low V_{π} (sub volt levels). The design and construction of the integrated-optics modulators using thin-film lithium niobate (TFLN) on a silicon-dioxide substrate is described. The optical waveguide geometry is optimized to provide low optical loss while maintaining strong coupling with the radio-frequency signal that drives the electro-optic phase modulation. This is done through a unique tapered waveguide design in which the optical waveguide width changes adiabatically from the single-mode width (300 nm) outside the electrode to a wider (600 nm) geometry through the electrode to enhance the electro-optic coupling and back. This unique design and platform yields waveguide-based EOMs with low driving voltage – characterized by V_{π} , which is the electrical voltage required to impose a π phase shift to the optical field, for the visible to near infrared spectral domain. The performance of the EOMs is shown in Fig. 2 of the manuscript. The measured V_{π} is given as a function of wavelength for one device at a modulation frequency of 1 MHz (all below 1 volt). The measured electronic-driving frequency response of the device is shown to have nearly 35 GHz electronic bandwidth for 738 nm optical wavelength with a V_{π} near 1 volt. These results are compared against the state of the art in Fig. 2, which shows significant improvement over prior results in terms of modulation bandwidth and V_{π} .

To demonstrate the potential of this platform for optical applications the authors present two experiments. First, an electro-optic frequency comb is demonstrated at three different central wavelengths (638 nm, 738 nm and 838 nm) by sending a continuous-wave (CW) laser to an EOM driven by a 3 W, 30 GHz RF signal to produce optical sidebands. The spectrum of the resulting optical radiation is using an optical spectrum analyzer (OSA), which shows over 50 comb lines for each wavelength, with nearly equal amplitudes across the spectrum. Since the repetition rate of the output pulse train should equal the line spacing, which is equal to the RF modulation frequency of 30 GHz, it would be difficult to measure in the time domain (requiring time resolution of better than 30 picoseconds). Thus one cannot directly verify the coherence of the comb in the current system unless something such as spectral shearing interferometry is performed.

The second application involves the spectral shearing of a narrow bandwidth optical pulse (approximately 0.9 GHz bandwidth). The pulse is synchronized with the EOM driving signal, which consists of a 100 MHz sinusoidal voltage with $V \sim 20 V_{\pi}$. By aligning the optical pulse with the rising or falling linear temporal region of the driving voltage signal – as depicted in Fig. 4 (a) of the manuscript,

one can achieve spectral shearing of the central frequency of the optical pulse. Prior work with single-photon-level signals has shown such spectral shearing, but not beyond the original pulse bandwidth. In this work, the authors are able to shift the central frequency by ± 6.4 GHz, which is over 7 times the original spectral bandwidth of the pulse. Thus one could imagine using this device for spectrally multiplexing single-photon sources into multiple frequency bins!

I believe this is timely and highly impactful work that is worthy of publication. However, there are two points that should be addressed by the authors.

1) Optical loss of devices: Nearly all key performance parameters for the modulators are discussed. However, I think the authors should clarify the optical loss of their devices. For example, in the abstract and introduction several key performance parameters for the visible and near-infrared wavelength TFLN modulators are noted with numerical values given for the devices constructed. The loss is mentioned in the abstract, but without specifying its value – “we realize VNIR amplitude and phase modulators featuring $V\pi L$'s of sub-1 V-cm, low optical loss, and high bandwidth EO response.” Furthermore, in paragraph 4 of section 3.1, the insertion loss of the devices is mentioned, “Combining this simulated absorption loss with the measured splitter and waveguide propagation loss [30], we estimate the device insertion loss for a 1 cm MZM device with the smallest gap (2.5 μm) to be ~ 1.4 dB.” However, there is no mention of the measured device insertion loss. Basically, it would be very useful to provide the measured optical insertion loss for the devices used for the comb and frequency shift. This helps the reader understand the full device performance – particularly for quantum applications where loss budget is extremely important.

2) Spectral distribution of spectrally sheared pulses. In Fig. 4 (c), which depicts the spectrum of optical pulses before and after spectral shearing, shows the sheared pulses with ‘side-lobes’. It is proposed that these arise from light in the orthogonal polarization. However, the authors do not present measurements that support this hypothesis. This seems like a measurement that should be readily achieved by using optical polarization control at the input and polarization analyzers at the output of the modulator.

In addition, I have some further suggestions listed below to aid with the general presentation.

Specific Comments:

In the abstract and introduction several key performance parameters for the visible and near-infrared wavelength thin-film lithium niobate electro-optic modulators are noted with numerical values given for the devices constructed. However, one particularly critical parameter, the loss, is not given. I encourage the authors to provide this information up front and comment on how this could be improved.

In Fig. 1c caption, please note that you vary the width of the “gap” region in for both the 300 and 600 nm top-width waveguide sections as described in the text.

Figure 1 caption has a strange symbol in the parentheses “ultra-low drive voltages (j 1 V).” Should this be “ultra-low drive voltages (< 1 V)?”

There is no reference to Figure 1d in the main text. Please change this.

In section 2, please clarify what specifications lead to the constraint of having the top width of the TFLN waveguide to be 300 nm. Is this to ensure a symmetric mode profile or some other property?

Beginning of Section 3.1, Change “We fabricated...” to “We fabricate...” so that the tense is consistent. Similarly, in the second (and third) paragraph of the same section “We performed” instead of “We perform.” I guess, just please read through to ensure the tense is consistent throughout the manuscript.

Figure 2b – The label is confusing – what is meant by (left-blue)? Also, in the label indicate that this is for a Mach-Zehnder interferometer modulator (not a phase modulator). On the inset of Fig. 2 b, please label axes.

Fig. 2c label – what is meant by the dashed-lines denote the 3-dB bandwidth w.r.t 3 GHz?

End of second to last paragraph of Section 3.1 – it seems strange to say “amplification of RF loss” – “increase of RF loss” might sound better (amplification and loss are typically conjugates).

In Fig. 3a – c, please give add axes labels to the insets. Also give the estimated comb line spacing in the figure caption (it looks like they have about 30 GHz line spacing).

For Fig. 3 d, indicate in the caption for which central optical wavelength the data corresponds and clarify that what is plotted is the “Normalized Transmission” through a scanning Fabry-Perot interferometer used to determine the spectrum

I would suggest changing the term ‘linewidth’ to ‘bandwidth’ when discussing the spectral shear ‘beyond the linewidth of the pulse’. I am used to thinking of linewidth associated with a continuous-wave source rather than a pulsed source of light.

For the spectral shearing experiment – clarify the specifications of the particular device used – what was the gap size, modulator length and V_{π} ?

What is the central wavelength of the 1 ns duration pulses used in the spectral shearing experiments shown in Fig. 4? Please include this in the manuscript.

In section 5.2.2 – please clarify in the first sentence what spectrum you are measuring. Is this the spectrum of an optical field that transmits through the modulator or something else? If so, is it a continuous wave field centered about ω_0 ? Also in this section there is a typo (i 15 GHz)

For all commercial devices / equipment listed, please give further details of the specific device. For example in Section 5.2.3 – what amplitude modulator model is used from EOSpace?

Overall, I find the work to be of high quality and likely to make significant impacts in both quantum information science and technology as well as classical optical signal processing. The key advancements here involve the high bandwidth and low V_{π} achieved with the integrated EOMs on the TFLN platform. The manuscript is well written and cites relevant work within the field. I find that the methods employed are sound and the results are supported by measurements performed. Thus, I believe the work presented merits publication in Nature Communications after suitably addressing my concerns above.

Reviewer #2 (Remarks to the Author):

The novelty is not sufficient. Prior work using TFLN within visible spectrum can be found in the references [30-32]. In addition, the bandwidth of LN and its E-O coefficient are much lower than other Pockels effect based materials. The authors should compare the TFLN performance, including the V_{π} .

and the bandwidth, with other E-O modulation platforms, such as E-O polymers/BaTiO₃/2-D materials, and then readdress the information in Fig. 2 (d).

The frequency comb shown in Fig. 3 are not clear. The authors should fabricate additional devices to clean the impurity of the waveguide's spatial mode, specific in the shorter wavelengths, and then verify the statements.

Reply of the authors to the reviewers' comments

The comments by the reviewers are written in blue. The reply by the authors is written in red. The references in this document are listed at the end and do not correspond to the reference numbers in the original manuscript.

Reviewers' comments

Authors' response

Manuscript modified text

Manuscript retained text in black

Reviewers #1 comments:

I have carefully read through the manuscript entitled, "Sub-1 Volt and High-Bandwidth Visible to Near-Infrared Electro-Optic Modulators," by Dylan Renaud et al. The authors present experimental results that demonstrate high-bandwidth electro-optic modulators (EOMs) for visible and near-infrared light with a low V_π (sub volt levels). The design and construction of the integrated-optics modulators using thin-film lithium niobate (TFLN) on a silicon-dioxide substrate is described. The optical waveguide geometry is optimized to provide low optical loss while maintaining strong coupling with the radio-frequency signal that drives the electro-optic phase modulation. This is done through a unique tapered waveguide design in which the optical waveguide width changes adiabatically from the single-mode width (300 nm) outside the electrode to a wider (600 nm) geometry through the electrode to enhance the electro-optic coupling and back. This unique design and platform yields waveguide-based EOMs with low driving voltage – characterized by V_π , which is the electrical voltage required to impose a pi phase shift to the optical field, for the visible to near infrared spectral domain. The performance of the EOMs is shown in Fig. 2 of the manuscript. The measured V_π is given as a function of wavelength for one device at a modulation frequency of 1 MHz (all below 1 volt). The measured electronic-driving frequency response of the device is shown to have nearly 35 GHz electronic bandwidth for 738 nm optical wavelength with a V_π near 1 volt. These results are compared against the state of the art in Fig. 2, which shows significant improvement over prior results in terms of modulation bandwidth and V_π . To demonstrate the potential of this platform for optical applications the authors present two experiments. First, an electro-optic frequency comb is demonstrated at three different central wavelengths (638 nm, 738 nm and 838 nm) by sending a continuous-wave (CW) laser to an EOM driven by a 3 W, 30 GHz RF signal to produce optical sidebands. The spectrum of the resulting optical radiation is using an optical spectrum analyzer (OSA), which shows over 50 comb lines for each wavelength, with nearly equal amplitudes across the spectrum. Since the repetition rate of the output pulse train should equal the line spacing, which is equal to the RF modulation frequency of 30 GHz, it would be difficult to measure in the time domain (requiring time resolution of better than 30 picoseconds). Thus one cannot directly verify the coherence of the comb in the current system unless something such as spectral shearing interferometry is performed. The second application involves the spectral shearing of a narrow bandwidth optical pulse (approximately 0.9 GHz bandwidth). The pulse is synchronized with the EOM driving signal, which consists of a 100 MHz sinusoidal voltage with $V \sim 20 V_\pi$. By aligning the optical pulse with the rising or falling linear temporal region of the driving voltage signal – as depicted in Fig. 4 (a) of the manuscript, one can achieve spectral shearing of the central frequency of the optical pulse. Prior work with single-photon-level signals has shown such spectral shearing, but not beyond the original pulse bandwidth. In this work, the authors are able to shift the central frequency by +/- 6.4 GHz, which is over 7 times the original spectral bandwidth of the pulse. Thus one could imagine using this device for spectrally multiplexing single-photon sources into multiple frequency bins! I believe this is timely and highly impactful work that is worthy of publication. However, there are two points that should be addressed by the authors.

We thank the reviewer for reading our manuscript, and paying careful attention to the content of this work. The

synopsis accurately captures the results of our work, and the suggested improvements and points of clarification are deeply appreciated. We also appreciate the positive comments regarding our work, especially the following assessment: “I believe this is timely and highly impactful work that is worthy of publication.”

1. Optical loss of devices: Nearly all key performance parameters for the modulators are discussed. However, I think the authors should clarify the optical loss of their devices. For example, in the abstract and introduction several key performance parameters for the visible and near-infrared wavelength TFLN modulators are noted with numerical values given for the devices constructed. The loss is mentioned in the abstract, but without specifying its value – “we realize VNIR amplitude and phase modulators featuring $V_{\pi}L$'s of sub-1 V·cm, low optical loss, and high bandwidth EO response.” Furthermore, in paragraph 4 of section 3.1, the insertion loss of the devices is mentioned, “Combining this simulated absorption loss with the measured splitter and waveguide propagation loss [30], we estimate the device insertion loss for a 1 cm MZM device with the smallest gap ($2.5 \mu\text{m}$) to be ~ 1.4 dB.” However, there is no mention of the measured device insertion loss. Basically, it would be very useful to provide the measured optical insertion loss for the devices used for the comb and frequency shift. This helps the reader understand the full device performance – particularly for quantum applications where loss budget is extremely important.

We thank the reviewer for this comment and for their careful attention to the device performance. We agree with the reviewer’s assessment, and have decided to fabricate additional devices to directly characterize the on-chip loss of the modulator. To do so, we have fabricated $3 \mu\text{m}$ gap modulators (same gap as that used in devices for frequency comb and frequency shifting experiments) and varied the electrode length to extract the modulator loss (i.e. cutback method). The results are produced below. From this, we have extracted optical losses of approximately $0.7 \text{ dB/cm} \pm 0.2$.

By combining these results with our measured facet loss using lensed fibers ($\sim 7 \text{ dB/facet}$), we obtain a total device insertion loss of $\sim 15 \text{ dB}$ for a 1 cm, $3 \mu\text{m}$ gap modulator. We note that the loss is dominated by the facet loss, which can be significantly improved upon using already demonstrated designs such as tapered couplers ($\sim 1 \text{ dB/facet}$). We have modified both the abstract and manuscript text to include these numbers and ensure they are clearly presented to the reader, and placed the measured data in the supplementary. Additionally, we have added a reference to comment on how the total loss can be reduced by using low-loss facet couplers operating at visible wavelengths.

We have added the following text to the abstract:

...Our Mach-Zehnder modulators exhibit a $V_{\pi}L$ as low as $0.55 \text{ V}\cdot\text{cm}$ at 738 nm , **on-chip optical loss of $\sim 0.7 \text{ dB/cm}$** , and EO bandwidths in excess of 35 GHz ...

We have modified the text of paragraph 3, section 3.1 as follows, and included two additional references to compare our modulator loss (on-chip loss) with on-chip losses reported in other TFLN visible works. We have also added a sentence on how the facet coupling loss can be reduced using our recently demonstrated tapered fiber coupling in LN:

We extract the on-chip modulator loss by fabricating and measuring $3 \mu\text{m}$ gap modulators of varying length. From this

we obtain an on-chip loss of $\sim 0.7 \pm 0.2$ dB/cm (see supplementary for details). This value is over an order of magnitude smaller than that reported in other recent demonstrations for VNIR LN modulators [32, 37]. Including the lensed fiber-to-chip coupling loss (~ 7 dB/facet), the total device insertion loss comes to ~ 15 dB. We note that because the total device insertion loss is dominated by coupling loss (mismatch between the lensed fiber and rib waveguide mode), it can be reduced by nearly an order of magnitude using techniques such as tapered fiber coupling [38]

Finally, we have added the plot and text to the supplementary as shown below:

I. MEASUREMENT OF MODULATOR LOSS

We extract the modulator on-chip loss by fabricating an array of $3 \mu\text{m}$ gap modulators of varying electrode length, measuring the transmission as a function of length over the 710-740 nm range, and then fitting the result. The device on-chip transmission loss and resultant linear fit are shown in Fig. S1 below:

Fig. 1. **Modulator Loss via Cutback Method a**, On-chip transmission loss as a function of metal electrode length. Data is plotted relative to the linear fit y-intercept. A linear fit yields a loss of 0.7 dB/cm, with ± 0.2 error arising from the fiber-to-chip coupling efficiency variation.

From this, we obtain an on-chip loss of 0.7 ± 0.2 dB/cm. The error is derived from the variation of the fiber-to-chip coupling efficiency. The lensed fiber-to-chip coupling efficiency is measured to be approximately 7 dB/facet, and arises from the large modal mismatch between the lensed fiber and rib waveguide modes. Thus, the total device insertion loss of our $3 \mu\text{m}$ gap, 1 cm long modulator is ~ 15 dB.

2. Spectral distribution of spectrally sheared pulses. In Fig. 4 (c), which depicts the spectrum of optical pulses before and after spectral shearing, shows the sheared pulses with ‘side-lobes’. It is proposed that these arise from light in the orthogonal polarization. However, the authors do not present measurements that support this hypothesis. This seems like a measurement that should be readily achieved by using optical polarization control at the input and polarization analyzers at the output of the modulator.

We thank the reviewer for this comment. We have redone the measurement and in the process discovered an issue with our polarization control of light coupled into the device. Upon retaking the data, we were able to effectively suppress these side lobes via polarization. We have updated figure 4 with the new data. In addition we have modified the following paragraph to remove reference to the side lobes and updated the measured performance parameters to be consistent with the new measurement.

We observe a spectral shift of ± 6.6 GHz when locking the pulse to the rising or falling edge, respectively (figure 4c). $\sim 85\%$ of input power is shifted into the desired lobe, with our estimate limited by the finite extinction of the input optical pulses (~ 20 dB), which can be improved via gating the detected signal.

In addition, I have some further suggestions listed below to aid with the general presentation. Specific Comments:

1s. In the abstract and introduction several key performance parameters for the visible and near-infrared wavelength thin-film lithium niobate electro-optic modulators are noted with numerical values given for the devices constructed. However, one particularly critical parameter, the loss, is not given. I encourage the authors to provide this information up front and comment on how this could be improved.

We thank the reviewer for the suggestion. Please see above our response to point 1. "Optical loss of devices" which lists the measurements and changes we have implemented to address this.

2s. In Fig. 1c caption, please note that you vary the width of the "gap" region in for both the 300 and 600 nm top-width waveguide sections as described in the text.

We have modified the caption of figure 1c to reflect the fact that the electrode gap (distance between ground and signal pads) varies over the contact region (where GSG probes contact the electrodes) to the interaction region (where microwave and optical fields strongly interact), and back again. Note that this electrode tapering is implemented so that we can easily contact the electrodes with GSG probes. At the contact region, the gap is $\sim 10 \mu\text{m}$ wide, and therefore the waveguide negligibly interacts with the RF mode until after the gap tapers down to the interaction region. We have also modified the figure by labelling the GSG "contact" and electrode-waveguide "interaction" regions of the modulator. The text modification at the end of the figure caption is as follows:

The micrograph additionally shows the unbalanced amplitude modulator waveguides, along with the Y-splitters, probe contact region of the electrodes, and the electrode gap taper along the probe contact region to the interaction region.

3s. Figure 1 caption has a strange symbol in the parentheses “ultra-low drive voltages (? 1 V).” Should this be “ultra-low drive voltages (< 1 V)?”

We thank the reviewer for bringing this to our attention. The text has been modified per the reviewer’s correction.

...In the time domain, VNIR amplitude modulators with ultra-low drive voltages (< 1 V) can modulate continuous-wave optical inputs at CMOS voltages...

4s. There is no reference to Figure 1d in the main text. Please change this.

We thank the reviewer for pointing this out. The text has been modified to include a reference to sub-figure 1d at the conclusion of the first paragraph in section 2.

An optical micrograph of a fabricated 5 mm long amplitude modulator is provided in figure 1d.

5s. In section 2, please clarify what specifications lead to the constraint of having the top width of the TFLN waveguide to be 300 nm. Is this to ensure a symmetric mode profile or some other property?

We note here that the *physical* constraint of having the top width of the TFLN waveguide be 300 nm comes from the single mode operating condition (i.e. having the waveguide only support *fundamental* transverse-electric and magnetic modes). If the waveguide is made larger, higher-order modes are supported. In the current manuscript text, at the beginning of paragraph 1, section 2, the following text is currently included to explain this:

"Outside of the electrode region, the waveguides are designed to be single mode (support transverse-electric, TE₀₀, and transverse-magnetic, TM₀₀) at 740 nm. We choose this constraint to minimize excitation of higher-order modes, which can lead to a reduction in EO performance in the electrode region. Using finite-difference eigenmode simulations (Lumerical), the required waveguide top width for single mode operation is determined to be approximately 300 nm."

6s. Beginning of Section 3.1, Change “We fabricated...” to “We fabricate...” so that the tense is consistent. Similarly, in the second (and third) paragraph of the same section “We performed” instead of “We perform.” I guess, just please read through to ensure the tense is consistent throughout the manuscript.

RESPOND. We have made the following modifications to ensure the tense used is consistent throughout:

- (sec 3.1)...varying gap sizes experimentally evaluate their performance...
- (sec 3.1)...we fabricate 1 cm long Mach-Zehnder modulators...

7s. Figure 2b – The label is confusing – what is meant by (left-blue)? Also, in the label indicate that this is for a Mach-Zehnder interferometer modulator (not a phase modulator). On the inset of Fig. 2 b, please label axes.

We thank the reviewer for outlining points of confusion in this figure. We have removed the statement "left-blue" from the label for figure 2b, included the statement "MZM" (which is defined in the caption header as Mach-Zehnder Modulator) throughout the figure caption, and added the y- and x-axis labels to the inset figure of 2b.

Fig. 2. Ultra-low V_π visible wavelength Mach-Zehnder modulators (MZMs) with greater than 35 GHz bandwidth.

a, Experimental setup illustration and measured low-frequency (1 MHz) V_π for a 1 cm MZM with an electrode gap of 3 μm . Data shown corresponds to V_π at $\lambda = 532, 638, 738, 838,$ and 938 nm. Simulated V_π is shown by the solid line. **b** Simulated and measured V_π (1 MHz) MZM at $\lambda = 738$ nm for varied electrode gap. Inset shows a measured extinction ratio of ~ 21 dB for a 1 cm long, 3 μm gap MZM. **c**, Frequency dependence of V_π for a 1 cm long MZM. A 3-dB EO bandwidth of ~ 35 GHz is extracted from the response. The dashed-lines denote the 3-dB bandwidth w.r.t 3 GHz. **d**, Comparison of modulator figure of merit BW/V_π (ratio between 3-dB EO bandwidth and half-wave voltage V_π) between this work, state-of-the-art commercial LN modulators, previous VNIR thin-film LN modulators, and other VNIR modulator platforms. This work exhibits significantly higher BW/V_π values than all previously reported works. The dashed lines correspond to constant values of BW/V_π . Note that for fair comparison, we compare the reported V_π at < 1 GHz for all devices. The TFLN data points with cross annotations denote devices for which the reported BW was limited by the equipment used. Details of referenced works can be found in the supplementary.

8s. Fig. 2c label – what is meant by the dashed-lines denote the 3-dB bandwidth w.r.t 3 GHz?

By "dashed lines", we are referring to the gray, dashed lines indicating the measured 3-dB EO bandwidth of the amplitude modulator (MZM). When measuring the 3-dB bandwidth - i.e. the frequency range over which the RF power required to drive the amplitude modulator doubles - we take a starting value as our reference. In this work, we measure the 3-dB bandwidth with respect to a reference value at 3 GHz. This is done for reasons outlined in the main text (see section 3.1, paragraph 5), and is reproduced below: "A non-DC reference is chosen due to both the rapid roll-off originating from the CPW impedance mismatch, and the commonly observed instability in LN modulators at low frequencies due to photorefractive effects."

However, to help clear up any confusion, we have added the following text to the figure caption:

...A 3-dB EO bandwidth of ~ 35 GHz is extracted from the response. The bandwidth is measured with respect to a low frequency reference, here taken to be 3 GHz. The grey dashed-lines denote the 3-dB bandwidth w.r.t 3 GHz...

9s. End of second to last paragraph of Section 3.1 – it seems strange to say “amplification of RF loss” – “increase of RF loss” might sound better (amplification and loss are typically conjugates).

We thank the reviewer for emphasizing this confusing language. We have modified the text per the reviewer’s suggestion ...it can be improved upon by implementing capacitively loaded traveling wave electrodes to reduce current crowding and its associated increase of RF loss...

10s. In Fig. 3a – c, please give add axes labels to the insets. Also give the estimated comb line spacing in the figure caption (it looks like they have about 30 GHz line spacing).

We thank the reviewer for the suggested improvements to figure 3. We have added axis markers and labeled the corresponding values. Additionally, we have added dashed lines to the inset figures indicating the region of the comb from which we are magnifying the comb lines. The caption has also been modified to indicate the 30 GHz comb line spacing.

Fig. 3. **Integrated Visible to Near-Infrared Electro-optic Frequency Combs.** Tunable frequency combs operating at **a**, $\lambda = 638$, **b**, 738 nm, and **c**, 838 nm with 30 GHz spacing and more than 50 lines. Insets show magnified view of the comb lines. The asymmetry shown in the 638 nm and 738 nm combs is attributable to the waveguide supporting higher order modes at shorter wavelengths (see supplementary). **d**, Normalized 1st modulated sideband as a function of applied RF frequency showing comb tunability up to 40 GHz. The pump wavelength is 738 nm. Spectra are under-sampled due to the limited resolution of the spectrometer. Gaussian fits are provided to guide the eye.

11s. For Fig. 3 d, indicate in the caption for which central optical wavelength the data corresponds and clarify that what is plotted is the “Normalized Transmission” through a scanning Fabry-Perot interferometer used to determine the spectrum

We thank the reviewer for the comments regarding the figure 3d caption. We must clarify that the spectra provided in figure 3d are resolved from a Czerny-Turner spectrometer (Princeton Instruments), not a Fabry-Perot interferometer. This information is included in section 5.2.2.. We have corrected the previously incorrectly labeled y-axis for figure 3d. It now reads "normalized intensity". We have also modified the caption text to include the pump wavelength (please see the caption modification in the figure above).

12s. I would suggest changing the term ‘linewidth’ to ‘bandwidth’ when discussing the spectral shear ‘beyond the linewidth of the pulse’. I am used to thinking of linewidth associated with a continuous-wave source rather than a pulsed source of light.

We thank the reviewer for this comment. We agree that using the term ”bandwidth” may make our point more clear. We have updated the text accordingly. Please see below for all changes.

- In figure caption 1a:

Similarly, sub-volt phase modulators enable VNIR frequency comb generation and frequency shifting over multiple pulse bandwidths...

- In figure caption 4c:

...showing a shift of ± 6.6 GHz (over 7 times the pulse bandwidth) relative to the original spectrum with no RF tone applied...

- In section 3.3:

To the best of our knowledge this is the first demonstration of shearing optical pulses beyond their optical bandwidth.

- In the conclusion:

... and measured spectral shifting with shifts beyond the intrinsic bandwidth of the pulse.

13s. For the spectral shearing experiment – clarify the specifications of the particular device used – what was the gap size, modulator length and V_π ?

We have added the following text in section 3.3 to include the requested information:

Our TFLN device is a 1 cm long, 3 μm gap phase modulator with a $V_\pi \sim 1\text{V}$ at 100 MHz.

14s. What is the central wavelength of the 1 ns duration pulses used in the spectral shearing experiments shown in Fig. 4? Please include this in the manuscript.

We have added the following text in section 3.3 to include the requested information:

...1 ns duration square-shaped optical pulses ($\lambda = 737$ nm) to the rising or falling linear regime...

15s. In section 5.2.2 – please clarify in the first sentence what spectrum you are measuring. Is this the spectrum of an optical field that transmits through the modulator or something else? If so, is it a continuous wave field centered about ω_0 ? Also in this section there is a typo (15 GHz)

We thank the reviewer for indicating points of clarification. We have included the requested information in section 5.2.2 as shown below:

...a sinusoidal RF tone is applied at varying frequencies and the resulting optical spectrum is measured. The optical frequency spectrum of an MZM given an input CW optical carrier frequency of ω_0 , an applied RF tone at frequency ω_m and amplitude V_0 ...

...The frequency spectrum is measured using a home-built Fabry-Perot cavity (linewidth = 200 MHz) for lower frequencies (< 15 GHz) and an optical spectrum analyzer (OSA) or Czerny-Turner spectrometer (Princeton Instruments, SpectraPro HRS) for higher frequencies...

16s. For all commercial devices / equipment listed, please give further details of the specific device. For example in Section 5.2.3 – what amplitude modulator model is used from EOSpace?

We have addressed the reviewer’s request by adding in the following information regarding instrument models:

- polarization controller Thorlabs FPC560 - 5.2.1
- OZ-optics lensed fiber type TSMJ-3A-650-4/125-0.25-20-2-10-1 - 5.2.1
- GSG probe 40A-GSG-100-F - 5.2.1
- EOspace Modulator - AZ-AV5-40-PFA-PFA-737

Overall, I find the work to be of high quality and likely to make significant impacts in both quantum information science and technology as well as classical optical signal processing. The key advancements here involve the high bandwidth and low V_{π} achieved with the integrated EOMs on the TFLN platform. The manuscript is well written and cites relevant work within the field. I find that the methods employed are sound and the results are supported by measurements performed. Thus, I believe the work presented merits publication in Nature Communications after suitably addressing my concerns above.

We again thank the reviewer for their positive opinion of the manuscript and recommendation to publish in Nature Communications.

Reviewers #2 comments:

The novelty is not sufficient. Prior work using TFLN within visible spectrum can be found in the references [30-32].

We do not agree with the reviewers' assessment of novelty. Although we acknowledge that previous visible modulators have been demonstrated on TFLN, none have matched the performance (in terms of half wave voltage, loss, or bandwidth) demonstrated in this work. To clearly compare the performance of our devices with the cited works, we list their relevant performance specifications here:

	This Work	Ref 30	Ref 31	Ref 32
V_π (V)	0.42-0.85 (532-938 nm)	8 (850 nm)	4.2 (780 nm)	3.3 (532 nm)
3-dB Bandwidth (GHz)	35	10 [†]	2.7	25 [†]
On-Chip Loss (dB)	0.7 ± 0.2	N/A*	N/A*	7.3
E.R. (dB, range reported)	21-25	15-30	19.4-27	23

†: Measured bandwidth limited by equipment.

*: Manuscript does not report a measured modulator loss.

Additionally, since the initial submission of this work, another VNIR TFLN modulator work has appeared on ArXiv. As shown below, the performance of our modulator still exceeds that reported in this new work:

	This Work	P. Sund (2022)
V_π (V)	0.42-0.85 (532-938 nm)	4.5 (940 nm)
3-dB Bandwidth (GHz)	35	6.5
On-Chip Loss (dB)	0.7 ± 0.2	0.8 ± 0.3
E.R. (dB, range reported)	21-25	21

We have added this work to both Figure 2d and the references list. Additionally, to make clear the improvement of our device relative to previous TFLN VNIR modulator works, we have added the following text to the caption of Figure 2: ...This work exhibits significantly higher BW/V_π values than all previously reported works, **including previously demonstrated TFLN VNIR modulators** [30-32, 37]...

And we have added the following table to section 1 of the supplementary to explicitly compare VNIR TFLN modulator performances:

	This Work	Ref 30	Ref 31	Ref 32	Ref 37
V_π (V)	0.42-0.85 (532-938 nm)	8 (850 nm)	4.2 (780 nm)	3.3 (532 nm)	4.5 (940 nm)
3-dB Bandwidth (GHz)	35	10 [†]	2.7	25 [†]	6.5
On-Chip Loss (dB)	0.7 ± 0.2	N/A*	N/A*	7.3	0.8 ± 0.3
E.R. (dB, range reported)	21-25	15-30	19.4-27	23	21

†: Measured bandwidth limited by equipment.

*: Manuscript does not report a measured modulator loss.

Beyond our superior device performance, we also emphasize that the novelty of our work stems from the fact that we use these previously unachieved performance metrics to enable two state-of-the-art demonstrations: EO VNIR frequency combs and EO frequency shearing of pulsed light beyond the input pulse bandwidth (7x bandwidth).

In addition, the bandwidth of LN and its E-O coefficient are much lower than other Pockels effect based materials. The authors should compare the TFLN performance, including the V-pi.L and the bandwidth, with other E-O modulation platforms, such as E-O polymers/BaTiO₃/2-D materials, and then readdress the information in Fig. 2 (d).

We thank the reviewer for this comment. Firstly, we emphasize that this work is solely focusing on modulators operating in the visible to near-IR (~ 500 - 1000 nm) range. While it is true that there are certain platforms that possess larger EO coefficients (e.g. BTO, organic polymers), modulators using these platforms have little to no demonstrations at visible wavelengths. Since the initial preparation of this manuscript, there has been a demonstration of a visible polymer-based modulator with an impressively low half-wave voltage (0.52 V), but it has a demonstrated bandwidth only 1 kHz, nearly 7 orders of magnitude lower than what we report [1]. Nonetheless, we have modified the manuscript to include this comparison in the main text.

Finally, we note that while other visible modulator platforms have recently been demonstrated [1, 2], they are presently limited to operating frequencies in the 1 - 100 KHz range.

For BaTiO₃, while there are some recent works exploring the possibilities of integrated electro-optics with BaTiO₃ with impressive half wave voltages [3, 4], these demonstrations have been limited to telecommunications wavelengths (e.g. BTO on silicon) as opposed to the visible wavelengths we are focusing on within this work. For 2D-Materials, device-level demonstrations, particularly at visible wavelengths, are limited [5]. To the best of our knowledge, only a single work demonstrating visible EO modulators based on 2D materials exists [6]. Furthermore, the performance of this device is orders of magnitude below the platforms that we compare in Fig. 2d. We emphasize to the reviewer that while we agree there are other 2D modulators with impressive performance, they do not operate at visible wavelengths but rather telecommunications wavelengths [7–9]. For certain 2D materials, the telecommunication operation stems from the fact that they have to be operated below the excitonic resonance, which is in VNIR region (e.g. ~ 730 nm for WSe₂) [9]. For these reasons, 2D material based modulators are not suitable for comparing with this work.

The frequency comb shown in Fig. 3 are not clear. The authors should fabricate additional devices to clean the impurity of the waveguide's spatial mode, specific in the shorter wavelengths, and then verify the statements.

We thank the reviewer for the comment. We note that the asymmetry in the spectra at shorter wavelengths are due to the waveguide supporting higher order modes, as well as the limited resolution of our optical spectrum analyzer at shorter wavelengths. We support our statements concerning the asymmetry of the comb by theoretical calculations (please see supplementary section 2) which agree well with our measurements. Despite these non-ideal conditions, we believe the presence of a periodic frequency comb is clear from the data presented.

We note that for a particular application where a flatter frequency spectrum is desired at a particular wavelength, a single mode filter section of the waveguide can be added straightforwardly as the manuscript suggests, although actually demonstrating this is beyond the scope of this current work.

-
- [1] S. Kamada, R. Ueda, C. Yamada, K. Tanaka, T. Yamada, and A. Otomo, Superiorly low half-wave voltage electro-optic polymer modulator for visible photonics, *Optics Express* **30**, 19771 (2022).
- [2] G. Liang, H. Huang, A. Mohanty, M. C. Shin, X. Ji, M. J. Carter, S. Shrestha, M. Lipson, and N. Yu, Robust, efficient, micrometre-scale phase modulators at visible wavelengths, *Nature Photonics* **15**, 908 (2021).
- [3] L. Czornomaz and S. Abel, Bto-enhanced silicon photonics—a scalable pic platform with ultra-efficient electro-optical modulation (Optica Publishing Group, 2022) pp. Th1J–1.
- [4] F. Eltes, C. Mai, D. Caimi, M. Kroh, Y. Popoff, G. Winzer, D. Petousi, S. Lischke, J. E. Ortmann, L. Czornomaz, *et al.*, A batio 3-based electro-optic pockels modulator monolithically integrated on an advanced silicon photonics platform, *Journal of Lightwave Technology* **37**, 1456 (2019).

- [5] X. Gan, D. Englund, D. Van Thourhout, and J. Zhao, 2d materials-enabled optical modulators: From visible to terahertz spectral range, *Applied Physics Reviews* **9**, 021302 (2022).
- [6] B. Li, S. Zu, J. Zhou, Q. Jiang, B. Du, H. Shan, Y. Luo, Z. Liu, X. Zhu, and Z. Fang, Single-nanoparticle plasmonic electro-optic modulator based on mos2 monolayers, *ACS nano* **11**, 9720 (2017).
- [7] C. T. Phare, Y.-H. Daniel Lee, J. Cardenas, and M. Lipson, Graphene electro-optic modulator with 30 ghz bandwidth, *Nature Photonics* **9**, 511 (2015).
- [8] M. Liu, X. Yin, E. Ulin-Avila, B. Geng, T. Zentgraf, L. Ju, F. Wang, and X. Zhang, A graphene-based broadband optical modulator, *Nature* **474**, 64 (2011).
- [9] I. Datta, S. H. Chae, G. R. Bhatt, M. A. Tadayon, B. Li, Y. Yu, C. Park, J. Park, L. Cao, D. Basov, et al., Low-loss composite photonic platform based on 2d semiconductor monolayers, *Nature Photonics* **14**, 256 (2020).

REVIEWERS' COMMENTS

Reviewer #1 (Remarks to the Author):

Dear Authors and Editor,

The authors have addressed all of my comments satisfactorily. Indeed, the changes implemented have greatly improved the quality of the manuscript. Specifically, the low intrinsic loss of this platform coupled with the high bandwidth and low v - π make the work stand out as a promising platform for quantum applications. Even though other systems such as polymer modulators have higher bandwidth and lower v - π , these also suffer from greater loss.

I am happy to strongly recommend this manuscript for publication in Nature Communications.

Reply of the authors to the reviewers' comments

Round 2:

Reviewers #1 comments:

Dear Authors and Editor, The authors have addressed all of my comments satisfactorily. Indeed, the changes implemented have greatly improved the quality of the manuscript. Specifically, the low intrinsic loss of this platform coupled with the high bandwidth and low v - π make the work stand out as a promising platform for quantum applications. Even though other systems such as polymer modulators have higher bandwidth and lower v - π , these also suffer from greater loss. I am happy to strongly recommend this manuscript for publication in Nature Communications.

We thank the reviewer for both their recommendation for publication and insightful comments during the peer review process.

Reviewers #2 comments: None